

# Long non-coding RNA polymorphisms on 8q24 are associated with the prognosis of gastric cancer in a Chinese population

Yangyu Zhang[1,*], Yanhua Wu[1,*], Zhifang Jia[1], Donghui Cao[1], Na Yang[1], Yueqi Wang[1], Xueyuan Cao[2] and Jing Jiang[1]

[1] Division of Clinical Research, First Hospital of Jilin University, Changchun, Jilin, China
[2] Department of Gastric and Colorectal Surgery, First Hospital of Jilin University, Changchun, Jilin, China
* These authors contributed equally to this work.

## ABSTRACT

**Background:** Gastric cancer (GC) remains the third leading cause of cancer death in China. Although genome-wide association studies have identified the association between several single nucleotide polymorphisms (SNPs) on 8q24 and the risk of GC, the role of these SNPs in the prognosis of GC in Chinese populations has not yet been fully evaluated. Therefore, this study was conducted to explore the association between long non-coding RNA (lncRNA) polymorphisms on 8q24 and the prognosis of GC.

**Methods:** We genotyped 726 surgically resected GC patients to explore the association between eight SNPs in the lncRNAs CCAT1 (rs10087719, rs7816475), PCAT1 (rs1026411), PRNCR1 (rs12682421, rs13252298), and CASC8 (rs1562430, rs4871789, rs6983267) transcribed from the 8q24 locus and the prognosis of GC in a Chinese population.

**Results:** We found that the patients carrying rs12682421 AA genotypes survived for a shorter time than those with the GG/GA genotype (HR = 1.39, 95% confidence interval (CI) [1.09–1.78]). Compared with the CC/CT genotype, the TT genotype of rs1562430 was associated with an increased risk of death (HR = 1.38, 95% CI [1.06–1.80]). Furthermore, the results also identified the rs1026411 SNP as an independent prognostic factor for poor survival in GC patients. Patients carrying AA/AG variant genotypes had a 36% increased risk of death compared to those carrying the GG genotype (HR = 1.36, 95% CI [1.06–1.74]). These findings suggested that the rs12682421, rs1026411 and rs1562430 SNPs may contribute to the survival of GC and be prognostic markers for GC.

## INTRODUCTION

Gastric cancer (GC) is the fifth most common malignancy and the third leading cause of cancer-related mortality in the world. A total of 1,033,701 new cases of GC were estimated to have occurred in 2018 (*Bray et al., 2018*). Although the incidence of GC has been declining in the last decades in most regions, it remains a common cancer among many

Corresponding authors
Xueyuan Cao, caoxy@aliyun.com
Jing Jiang,
jiangjing19702000@jlu.edu.cn

populations in East Asia. Due to a high incidence rate and a large population, more than 40% of GC cases worldwide have occurred in China, according to GLOBOCAN 2012 (*Ferlay et al., 2015*). In the last decade, the mortality rate of GC has declined conspicuously due to the improved treatment approaches, but the prognosis of GC is still poor; the 5-year survival rate is 29.0% (*Zeichner et al., 2017*).

Regarding treatment approaches, tumorectomy with adjuvant or neoadjuvant chemotherapy and radiotherapy are the most effective treatments for GC. However, despite improvements in surgical and adjuvant multimodal treatments, the prognosis of GC is still poor due to late diagnosis and extreme intra- and inter-tumour heterogeneity (*Bonelli et al., 2019*). The heterogeneity makes the selection of treatment options difficult, and previous studies have found that patients with the same pathological stage and tumour grade who receive similar therapies may have different clinical outcomes, a finding that indicates the significance of individual variants influenced by genetic and environmental factors (*Wang et al., 2018*). Therefore, exploring genetic variations in key genes involved in tumour progression as biomarkers to improve the prognosis prediction of GC patients is imperative.

The results from genome-wide association studies (GWASs) have also identified single nucleotide polymorphisms (SNPs), the most common type of genetic variations in the human genome, in relation to the tumourigenesis of GC (*Abnet et al., 2010*; *Sakamoto et al., 2008*; *Shi et al., 2011*). GWASs have identified several loci, including 1q22, 5p13.1 and 8q24, that are associated with GC susceptibility, mainly in populations in Asia (*Saeki et al., 2013*; *Shi et al., 2011*; *Wadhwa et al., 2013*). Particularly, a series of evidence has suggested that 8q24 chromosome region can not only affect GC susceptibility (*Zhi, Shi & Liu, 2017*), but also confer GC patients with different prognosis (*Ma et al., 2015*; *Wang et al., 2016*), which further verified that genetic background play an important role in gastric carcinogenesis and progression.

Although paying attention to known genes might generate further understanding in development and therapy of GC, newly-developed markers such as long non-coding RNAs (lncRNAs) may lead novel insight into the mechanism of GC risk or treatment. LncRNAs are noncoding transcripts that are more than 200 nucleotides long. Although they were initially regarded as 'transcriptional noise', increasing studies have found that lncRNAs can regulate local or global gene expression through transcriptional, post-transcriptional and epigenetic regulation (*Mercer, Dinger & Mattick, 2009*). As lncRNAs play multiple roles in the regulation of gene expression, aberrant lncRNA expression may therefore occur during carcinogenesis and disease development. These advantages make lncRNAs potential biomarkers for the diagnosis, prognosis and therapy of a variety of cancers, including GC (*Wu & Hsieh, 2019*; *Yuan et al., 2018*; *Zhao et al., 2015*).

The 8q24 chromosome region has been reported to express several lncRNAs in different human tumours (*Huang, Zhang & Shao, 2018*; *Tong et al., 2018*; *Xiang et al., 2014*). The association between polymorphisms in lncRNAs and the risk of GC has been studied in several ethnicities (*Labrador et al., 2015*; *Pan et al., 2016*; *Ülger et al., 2017*). However, there are few studies that have investigated the prognostic value of lncRNA polymorphisms on 8q24 in GC patients. Hence, this study was performed to examine

whether variants of lncRNA colon cancer-associated transcript (CCAT1), prostate cancer-associated transcript 1 (PCAT1), prostate cancer non-coding RNA 1 (PRNCR1) and cancer susceptibility candidate 8 (CASC8) genes on chromosome 8q24 are associated with survival in a Chinese population with GC. It may bring benefits for individualised treatment and consequently improve survival outcomes.

## MATERIALS AND METHODS

### Study population

Subjects of the study were the newly diagnosed GC patients recruited from the Department of Gastric and Colorectal Surgery of the First Hospital of Jilin University from 2008 to 2013. A total of 756 patients who underwent tumourectomy without receiving chemotherapy or radiotherapy before surgery were enrolled in this study. The individual characteristics (gender, age) and clinical data (tumour size, histological type, histological grade, lymph metastasis, distant metastasis, depth of invasion, neural invasion and therapy) were collected from the medical records. TNM classification, based on the 2010 seventh edition of the American Joint Committee on Cancer (AJCC) guidelines (*Washington, 2010*), was used to evaluate the clinical stage of the cancer. The evaluation of *Helicobacter pylori* infection was performed via a serum immunoglobulin G (IgG) antibody test by an enzyme-linked immunosorbent assay (ELISA) using an *H. pylori*-IgG ELISA kit (Biohit, Helsinki, Finland). Postoperative chemotherapy was identified as an effective therapy for at least three cycles.

### Ethics statement

Each patient in this study signed an informed consent form before sample and information collection. This study was approved by the ethics committees of the First Hospital of Jilin University (2013-005).

### Follow-up

The follow-up of the patients was conducted 3 months, 6 months, and 1 year after surgery and every 1 year thereafter until the death of the patient or loss to follow-up. The data from each follow-up visit were collected. Subjects were excluded if they were lost to follow-up at the first phone interview or died due to complications of the surgery during the perioperative period (within 30 days after surgery). The survival time was considered as the duration (i) from the date of the surgery to the date of death if the GC patient had died or (ii) from the date of the surgery to the date of the last phone interview if the patient was lost to follow-up or to the end of the study if the patient was still alive.

### Tagging SNP selection

From whole blood sample of each patient, we extracted genomic DNA using a MagPure Tissue and Blood DNA KF Kit (Magen, Guangzhou, China). The tag SNPs and the well-studied SNPs on 8q24 that were previously reported be associated with gastrointestinal tumours were selected. These SNPs included CCAT1 rs10087719, CCAT1 rs7816475, PCAT1 rs1026411, PRNCR1 rs12682421, PRNCR1 rs13252298, CASC8

rs1562430, CASC8 rs4871789 and CASC8 rs6983267. The SNPinfo (http://snpinfo.niehs.nih.gov/), GVS (http://gvs.gs.washington.edu/GVS147/) and F-SNP (*Lee & Shatkay, 2008*) databases were used to select tag SNPs. The minor allele frequency of all the SNPs was > 0.05 based on the Han Chinese Population.

## Genotyping

Single nucleotide polymorphism genotyping was conducted by the MassARRAY technology platform (Sequenom, CA, USA) and was determined by the Bio Miao Biological Technology Co., Ltd. (Beijing, China). The detection rates for rs10087719, rs7816475, rs1026411, rs12682421, rs13252298, rs1562430, rs4871789 and rs6983267 were 100%, 98%, 99%, 100%, 100%, 100%, 98%, and 94%, respectively. The linkage disequilibrium (LD) was established with a threshold of the pairwise $r^2$ coefficient greater than 0.80 and the extent of LD between the eight SNPs were estimated using Haploview 4.2 (Broad Institute of MIT and Harvard, Cambridge, MA, USA). None of the eight SNPs were located at CCAT1, PRNCR1 or CASC8 with LD.

## Transcription factor binding site prediction

Transcription factor binding sites (TFBS) were predicted using the PROMO database (http://alggen.lsi.upc.es/cgi-bin/promo_v3/promo/promoinit.cgi?dirDB=TF_8.3).

## Statistical analysis

Frequency and proportion were used to describe the categorical variables. A goodness of fit $\chi^2$ test was used to test the Hardy-Weinberg equilibrium (HWE) of each SNP. Survival curves of the GC patients based on each SNP were plotted by the Kaplan–Meier method and were compared by log-rank test. The Cox regression model was used to calculate hazard ratios (HRs) with 95% confidence intervals (CIs) and to evaluate the associations between genotypes of each SNP and overall survival after adjusting for potential confounders (age, gender, *H. pylori*, tumour size, TNM stage, histological type, histological grade, chemotherapy, lymph vascular invasion and neural invasion). We used the Bonferroni correction method to adjust for multiple testing with a significance threshold set at $P = 0.00625$ (0.05/8 SNPs). All statistical analyses were performed using the SPSS 21.0 software (IBM SPSS, IBM Corp, Armonk, NY, USA).

# RESULTS

## Characteristics of patients

A total of 756 diagnosed GC patients were enrolled in the study. Fourteen patients died of complications from the surgery during the preoperative period, seven patients were lost to follow-up at the first phone interview and the genotyping of nine patients failed. The remaining 726 patients were included in the study for the subsequent analysis. At the end of the study, 27 patients were lost to follow-up, 357 patients died, and 342 patients were alive (Fig. 1). The duration of follow-up was from 1 month to 109 months, and

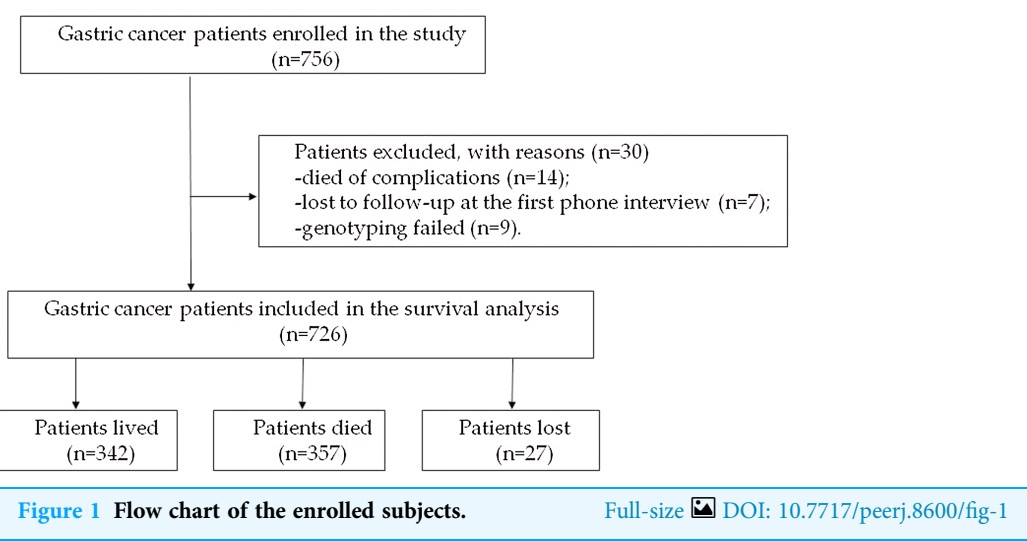

**Figure 1  Flow chart of the enrolled subjects.**     

the median follow-up time was 70.7 months. The characteristics of the 726 patients are shown in Table 1.

## Genotype and allele frequencies of the eights SNPs

The genotype frequencies of seven SNPs (rs10087719, rs7816475, rs1026411, rs12682421, rs13252298, rs1562430, rs4871789) in the subjects were in HWE with non-significant $\chi^2$ values ($P > 0.05$). The rs6983267 locus, however, was found to deviate from the HWE ($P < 0.001$). We randomly selected 10% of the samples for repeat genotyping of rs6983267, but the result remained the same. The distributions of the genotype and allele frequencies of the eight SNPs in subjects are shown in Table 2. After Bonferroni correction for multiple testing, none of the eight SNPs were significantly associated with the survival of GC. However, there was a statistically significant tendency for rs12682421 (log-rank $P = 0.03$). As shown in Table 2, patients with the rs12682421 GA genotype tended to have a better prognosis (HR = 0.77, 95% CI [0.61–0.97]) than patients with the AA genotype.

## Multivariate Cox regression analysis of SNPs and gastric cancer survival

A multivariate stepwise Cox regression model was performed to explore the independent prognostic factor for GC. The results showed that tumour size, TNM stage, lymph vascular invasion and chemotherapy were associated with the survival of the patients. The AA genotypes of rs12682421 were associated with a significantly increased risk of death compared with that of the GG/GA genotype (HR = 1.39, 95% CI [1.09–1.78]). The TT genotypes of rs1562430 were also associated with increased risk of death compared with that of the CC/CT genotype (HR = 1.38, 95% CI [1.06–1.80]). The results also identified rs1026411 SNP as an independent prognostic factor for the poor survival of GC patients; patients carrying AA/AG genotypes had a 36% increased risk of death compared to those carrying the GG genotype (HR = 1.36, 95% CI [1.06–1.74]) (Table 3). The survival curves of the three SNPs are shown in Fig. 2.

**Table 1 Characteristics of the GC patients.**

| Variables | | N | % |
|---|---|---|---|
| Age (years) | <60 | 340 | 46.8 |
| | ≥60 | 386 | 53.2 |
| Gender | Male | 547 | 75.3 |
| | Female | 179 | 24.7 |
| Smoking | Yes | 281 | 39.0 |
| | No | 440 | 61.0 |
| Drinking | Yes | 198 | 27.4 |
| | No | 524 | 72.6 |
| Family history | Yes | 47 | 6.5 |
| | No | 672 | 93.5 |
| *H. pylori* | Positive | 438 | 68.7 |
| | Negative | 200 | 31.3 |
| Tumour size | <5 cm | 420 | 57.9 |
| | ≥5 cm | 306 | 42.1 |
| TNM stage | I | 129 | 17.8 |
| | II | 271 | 37.3 |
| | III | 326 | 44.9 |
| Histological type | Tubular | 571 | 78.6 |
| | Signet-ring cell | 68 | 9.4 |
| | Other | 87 | 12.0 |
| Histological grade | Low-grade | 218 | 30.0 |
| | High-grade | 508 | 70.0 |
| Lymph vascular invasion | Yes | 514 | 70.8 |
| | No | 212 | 29.2 |
| Neural invasion | Yes | 399 | 55.0 |
| | No | 327 | 45.0 |
| Chemotherapy | Yes | 314 | 43.3 |
| | No | 412 | 56.7 |

## Stratified analysis of the genotypes associated with gastric cancer prognosis

Moreover, the associations between the three SNPs (rs12682421, rs1562430, rs1026411) and the survival of the GC patients was evaluated by a stratified analysis of tumour size, TNM stage, lymph vascular invasion and chemotherapy. Compared to patients with the GA/GG genotype of rs12682421, patients with the AA variant genotype had a higher death risk in the subgroup of patients with tumour sizes <5 cm (HR = 1.59, 95% CI [1.11–2.28]), an advanced TNM stage (TNM stage II: HR = 1.68, 95% CI [1.03–2.76]; TNM stage III: HR = 1.41, 95%CI [1.05–1.90]), lymph vascular invasion (HR = 1.49, 95% CI [1.15–1.95]) or no postoperative chemotherapy(HR = 1.40, 95% CI [1.01–1.94]) (Fig. 3A). Compared to patients with the CC/CT genotype, patients carrying TT genotypes of rs1562430 had a higher death risk in the subgroup of patients with tumour sizes ≥5 cm

**Table 2 Distributions of the genotypes of the gastric cancer patients.**

| Gene | Genotypes | | Patients, N | Death, N (%) | MST | HR (95% CI) | Log rank P |
|------|-----------|---|-------------|--------------|-----|-------------|------------|
| CCAT1 | rs10087719 | AA | 495 | 250 (50.50) | 64.89 | 1.00 | 0.73 |
| | | AG | 209 | 99 (47.37) | 75.70 | 0.93 [0.74–1.18] | |
| | | GG | 20 | 8 (40.00) | 54.67* | 0.81 [0.40–1.63] | |
| CCAT1 | rs7816475 | GG | 572 | 280 (48.95) | 70.04 | 1.00 | 0.65 |
| | | AG | 134 | 69 (51.49) | 56.38 | 1.08 [0.83–1.41] | |
| | | AA | 6 | 2 (33.33) | 55.99* | 0.61 [0.15–2.45] | |
| PCAT1 | rs1026411 | GG | 257 | 120 (46.69) | 75.20 | 1.00 | 0.32 |
| | | AG | 353 | 181 (51.27) | 60.85 | 1.20 [0.95–1.50] | |
| | | AA | 111 | 53 (47.75) | 63.07* | 1.11 [0.80–1.53] | |
| PRNCR1 | rs12682421 | AA | 443 | 230 (51.92) | 58.51 | 1.00 | 0.03 |
| | | GA | 248 | 106 (42.74) | 68.84* | 0.77 [0.61–0.97] | |
| | | GG | 33 | 21 (63.64) | 37.78 | 1.25 [0.80–1.95] | |
| PRNCR1 | rs13252298 | AA | 330 | 158 (47.88) | 70.05 | 1.00 | 0.75 |
| | | AG | 315 | 157 (49.84) | 69.29 | 1.09 [0.87–1.36] | |
| | | GG | 79 | 42 (53.16) | 55.39 | 1.07 [0.76–1.51] | |
| CASC8 | rs1562430 | TT | 505 | 257 (50.89) | 64.30 | 1.00 | 0.17 |
| | | CT | 199 | 93 (46.73) | 79.05 | 1.13 [0.84–1.53] | |
| | | CC | 19 | 6 (31.58) | 75.20 | 0.68 [0.35–1.16] | |
| CASC8 | rs4871789 | GG | 251 | 129 (51.39) | 64.89 | 1.00 | 0.64 |
| | | AG | 344 | 165 (47.97) | 73.50 | 0.90 [0.71–1.13] | |
| | | AA | 119 | 58 (48.74) | 67.35 | 0.91 [0.67–1.24] | |
| CASC8 | rs6983267 | TT | 172 | 89 (51.74) | 65.68 | 1.00 | 0.59 |
| | | GT | 416 | 195 (46.88) | 79.05 | 1.01 [0.82–1.24] | |
| | | GG | 95 | 48 (50.53) | 67.35 | 0.93 [0.84–1.09] | |

**Notes:**

MST, median survival time, months.

* Mean survival time was provided when MST could not be calculated.

**Table 3 Stepwise Cox regression analysis of gastric cancer survival.** Age, gender, *H. pylori*, tumour size, TNM stage, histological type, histological grade, chemotherapy, lymph vascular invasion, neural invasion and eight SNPs polymorphism were used as variables in the regression model.

| Genotypes | P | Adjusted HR (95% CI) |
|-----------|---|----------------------|
| Tumour size | 0.005 | 1.42 [1.11–1.81] |
| Lymph vascular invasion | <0.001 | 2.09 [1.45–3.02] |
| TNM stage | | |
|   II vs. I | 0.007 | 2.25 [1.24–4.07] |
|   III vs. I | <0.001 | 6.83 [3.77–12.36] |
| Chemotherapy | 0.007 | 0.72 [0.56–0.91] |
| rs12682421 (AA vs. GA+GG) | 0.009 | 1.39 [1.09–1.78] |
| rs1562430 (TT vs. CC+CT) | 0.016 | 1.38 [1.06–1.80] |
| rs1026411 (AA+AG vs. GG) | 0.017 | 1.36 [1.06–1.74] |

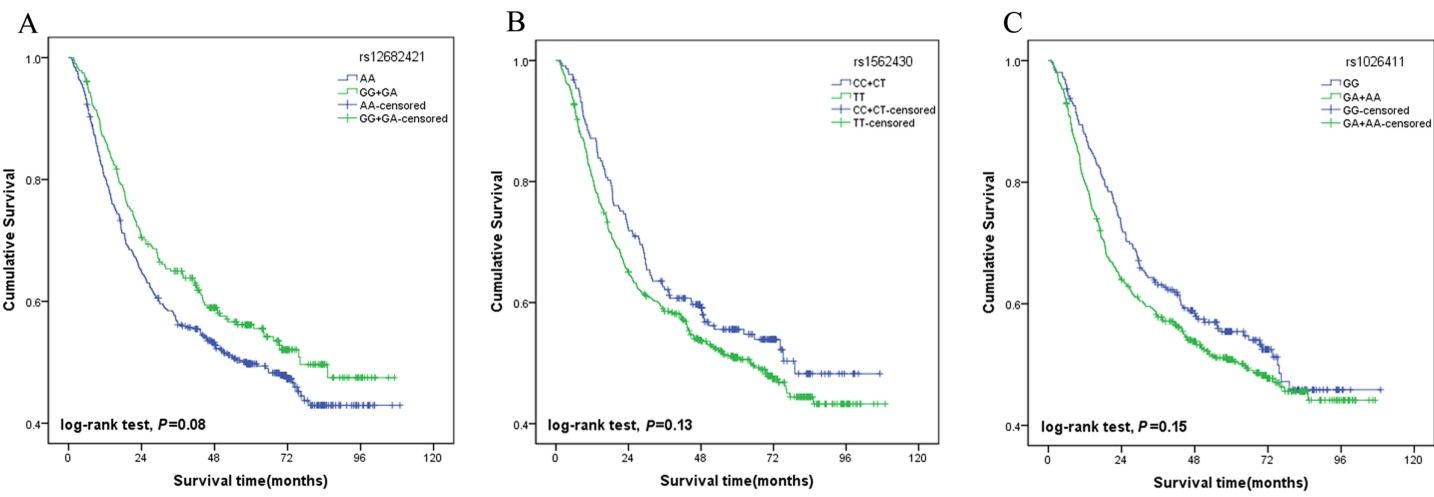

**Figure 2 Association of genotypes with overall survival in gastric cancer patients.** (A) Plot for rs12682421 using the dominant model (GG/GA vs. AA); (B) Plot for rs1562430 using the dominant model (CC/CT vs. TT); (C) Plot for rs1026411 using the dominant model (AA/AG vs. GG).

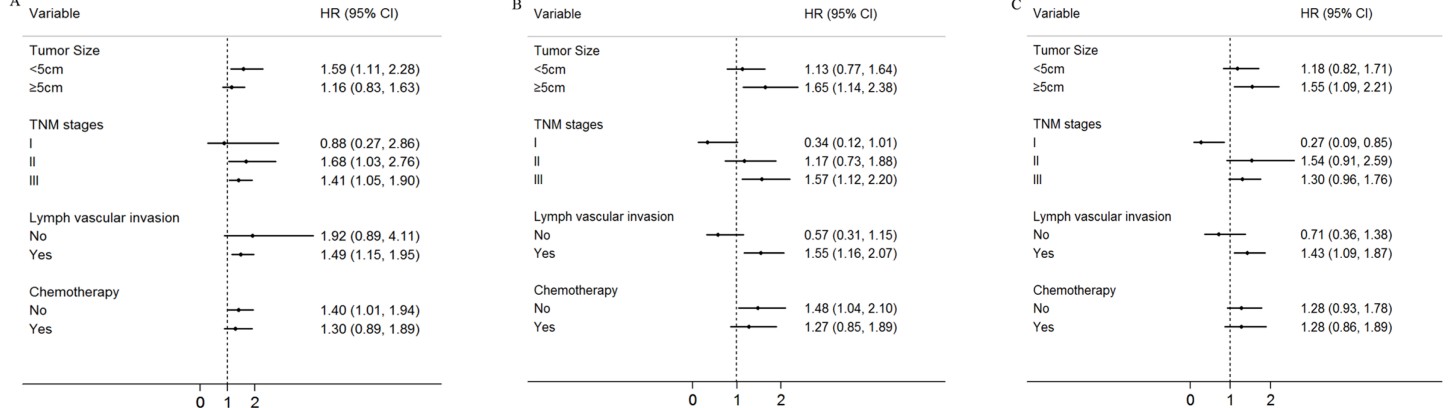

**Figure 3 Stratified analysis for rs12682421, rs1562430 and rs1026411genotypes associated with gastric cancer patients' survival.** Age, gender, *H. pylori*, tumour size, TNM stage, histological type, histological grade, chemotherapy, lymph vascular invasion, neural invasion and eight SNP polymorphisms were used as variables in the regression model. (A) Stratified analysis of rs12682421 genotypes associated with gastric cancer patients' survival (AA vs. GA+GG); (B) Stratified analysis of rs1562430 genotypes associated with gastric cancer patients' survival (TT vs. CC+CT); (C) Stratified analysis of rs1026411 genotypes associated with gastric cancer patients' survival (AA+AG vs. GG).

(HR = 1.65, 95% CI [1.14–2.38]), TNM stage III (HR = 1.57, 95% CI [1.12–2.20]), or lymph vascular invasion (HR = 1.55, 95%CI [1.16–2.07]) and in the subgroup of patients who did not receive postoperative chemotherapy (HR = 1.48, 95% CI [1.04–2.10]) (Fig. 3B). Compared to patient with the GG genotype of rs1026411, patients with the AA/AG variant genotype had a higher death risk in the subgroup of patients with tumour sizes ≥5 cm (HR = 1.55, 95% CI [1.09–2.21]) or lymph vascular invasion (HR = 1.43, 95% CI [1.09–1.87]) (Fig. 3C).

## DISCUSSION

Clinically, therapeutic decision making and prognostic prediction for GC patients still depend on the TNM staging system. However, due to the significant heterogeneity within the same stage, the TNM stage alone is not sufficient to predict the prognosis of GC. More significantly, although the TNM staging system classifies patients into subgroups with different clinical outcomes, it provides limited information about therapeutic effects in individual patients (*Li et al., 2017*; *Qu et al., 2015*). Therefore, it is crucial and necessary to identify new biomarkers for GC patients to complement the TNM staging system in order to improve the prediction of prognosis and guide therapeutic decisions. Our study focused on the investigation of eight lncRNA SNPs on 8q24 that predispose GC patients to survival. Multivariate analysis revealed that the AA genotype of rs12682421, the TT genotype of rs1562430 and the AA/AG genotype of rs1026411 could serve as potential markers to predict the unfavourable survival of GC patients in the Chinese population. Furthermore, the prominent prognostic effect of the three SNPs was more evident in advanced subgroups of GC patients.

Recent research has suggested that lncRNAs, as oncogenes or tumour suppressor genes, could be involved in the development of cancer and be associated with tumour metastasis and prognosis (*Song et al., 2017*; *Weidle et al., 2017*). Given that the majority of the GWAS-identified cancer risk SNPs are located in the noncoding region, the expression and function of lncRNAs are more likely to be impacted by the SNPs (*Gao & Wei, 2017*). Moreover, GWAS have identified 8q24 as a hotspot for cancer-associated SNPs on account of the strength, density and high allele frequency of these SNPs (*Sur et al., 2013*). Although several studies have revealed that lncRNAs on 8q24, including PRNCR1, CCAT2 and PCAT1, encompass the cancer predisposition SNPs (*Guo et al., 2019*; *Ling et al., 2013*; *Wang & Wang, 2019*; *Yang et al., 2019*), the prognostic significance of these lncRNAs in GC patients has not yet been fully explored.

Prostate cancer non-coding RNA 1, a lncRNA transcribed from 8q24, participates in the carcinogenesis of prostate cancer (PCa) by activating androgen receptor (AR) (*Chung et al., 2011*); in addition, polymorphisms of the lncRNA PRNCR1 were noted in many cancers, including colorectal cancer (*AlMutairi & Parine, 2019*), prostate cancer (*Sattarifard et al., 2017*), and GC (*He et al., 2017*). A meta-analysis conducted by *Huang, Zhang & Shao (2018)* showed that rs16901946 of PRNCR1, which was in complete LD with rs12682421, was associated with an increased risk of GC in the dominant model. A study performed by *He et al. (2017)* that aimed to assess the GC susceptibility and GC prognostic value of the polymorphisms in PRNCR1, found that rs16901946 G allele carriers (linked with rs12682421 G allele) have an increased risk of GC, but this polymorphism did not exhibit any significant prognostic value for GC. However, in our study, we found that compared with the GA/GG genotype, the PRNCR1 rs12682421 AA genotype was a poor prognostic factor for GC, which is different from the results of the study by He et al. We considered that the reasons for the inconsistent conclusion may be due to the fact that doctor He's study has a smaller sample size ($N = 494$), a shorter
follow-up time (the patients were followed for up to 4 years) and fewer events compared with our study.

Rs1562430 is located in the intron of CASC8, a long noncoding RNA (lncRNA), and overlaps with the POU5F1B gene. Previous studies have revealed that the rs1562430 SNP has a strong association with the risk of breast cancer and colorectal cancer (*He et al., 2011*; *Kim et al., 2012*; *Silvestri et al., 2015*). *Ma et al. (2015)* found that there were no associations between the rs1562430 genotype and the survival of GC patients in a Chinese population. However, in the present study, we found that rs1562430 TT was associated with a significantly lower survival rate in GC patients than CC/CT. The study conducted by Ma et al. included patients with TNM stage IV, and the histological type of 42.5% of patients was intestinal. Conversely, our study excluded patients with TNM stage IV, and more than 70% of patients exhibited an intestinal histological type. Moreover, nearly 70% of the patients in our study were infected with *H. pylori*. Therefore, the differences in the pathogenic environment and genetic background of the patients included in the two studies may be the reasons for the inconsistent results.

Existing studies have found that PCAT1 overexpression occurred in PCa, lung cancer and colorectal cancer (*Ge et al., 2013*; *Prensner et al., 2011*; *Zhao, Hou & Zhan, 2015*). *Shi et al. (2015)* identified that ESCC patients with high levels of PCAT1 had poorer survival times than those with low levels of PCAT1. Moreover, recent studies have suggested that a PCAT1 genetic variant may play an essential role in the susceptibility to several cancers (*Ren et al., 2017*; *Zhao, Hou & Zhan, 2015*). *Yuan et al. (2018)* found that rs1902432 in PCAT1 was significantly associated with an increased risk of PCa, and *Lin et al. (2017)* found that rs710886 of PCAT1 was significantly associated with bladder cancer risk in a Chinese population. As far as we know, no study has been conducted on the role of PCAT1 polymorphisms in the prognosis of GC. In the present study, we found that, compared to GG, rs1026411 AA/AG was associated with a poor prognosis of GC patients (HR = 1.33, 95% CI [1.03–1.70]).

Previous studies have found that SNPs in lncRNAs have different prognostic values when they occur with different clinical features (*Ma et al., 2015*; *Xiong et al., 2017*); therefore, we conducted a stratified analysis by tumour size, TNM stage, lymph vascular invasion and chemotherapy. The unfavourable prognostic effects of rs1026411, rs1562430 and rs12682421 were more evident among patients with increased TNM stage and lymph vascular invasion, which indicated that these three SNPs may have higher predictive value in advanced stages of GC. Previous studies have demonstrated that genomic polymorphisms can affect drug transport, metabolism and cellular response and cause individual variations in terms of the response and even overall survival (*Ulrich, Robien & McLeod, 2003*). Increasing evidence has also suggested that SNPs in some lncRNAs are related to chemotherapy response and could provide effective therapeutic targets for GC treatment (*Ozgur et al., 2019*; *Zhang & Du, 2016*). Our results showed that in patients who did not receive chemotherapy, the rs12682421 AA genotype and rs1562430 TT genotype could predict poor survival (HR = 1.40, 95% CI [1.01–1.94]; HR = 1.48, 95% CI [1.04–2.10]); however, in the subgroup of patients who received chemotherapy, the difference in the genomic polymorphisms of the two SNPs disappeared. The effect of the

genetic background of the two SNPs may be overshadowed by the advantages of chemotherapy, considering that the multivariate analysis results showed that patients with postoperative chemotherapy tended to have a favourable prognosis (HR = 0.72, 95% CI [0.56–0.91]).

Because lncRNAs could play a crucial role in the regulation of gene expression via transcription and transcription factors often play important roles in tumourigenesis, we used bioinformatics data from the PROMO TFBS database to predict the possible functions of rs12682421, rs1562430 and rs1026411. We found that the C allele at the rs1562430 locus allowed binding to the glucocorticoid receptor α (GRα), which is a transcription factor that increases genes that participate in cell cycle arrest and apoptosis (*Kumar, Johnson & Thompson, 2004*; *Patki & McFall, 2018*; *Yemelyanov et al., 2006*). GRα could be bound and activated by glucocorticoids, and previous studies have shown that a higher expression of glucocorticoids receptors has been correlated with a better prognosis in bladder cancer (*Ishiguro et al., 2014*; *Zheng et al., 2012*). Therefore, we considered that the rs1562430 T allele may be associated with a lower GRα expression, which affects the prognosis of the GC patients. Additionally, the A allele at the rs12682421 locus was found to be allowed binding to the GRβ transcription factor, which is a different isoform of GR. GRβ lacks the ligand-binding domain for glucocorticoids (*Hinds et al., 2010*) and has been indicated to inhibit GRα (*Leung et al., 1997*; *Hinds et al., 2010*; *Kubin et al., 2016*). GRβ has been demonstrated to be involved in the migration of bladder cancer and brain cancer (*Mcbeth et al., 2016*; *Ying et al., 2013*), and some other studies have also reported that GRβ levels are elevated in inflammatory diseases and cancers, leading to increased progression (*Zhu et al., 2007*; *Marino et al., 2016*; *Psarra et al., 2005*). Hence, we hypothesise that the poor prognosis of patients with the rs12682421 A allele may be associated with a higher GRβ expression. Moreover, we found that the G allele at the rs1026411 locus facilitated binding to the polyomavirus enhancer activator 3 (PEA3) transcription factor, which belongs to the PEA3 subfamily within the E-twenty-six domain transcription factor superfamily (*Kandemir et al., 2017*). Members of the PEA3 subfamily have been demonstrated in previous studies be associated with a variety of cancers (*Cowden Dahl, Zeineldin & Hudson, 2007*; *Keld et al., 2010*; *Kim et al., 2015*), but a study conducted by *Keld et al. (2011)* showed that PEA3 upregulation in isolation does not predict prognosis in any stage of GC. The specific roles of GRα, GRβ and PEA3 in GC should be verified in further studies.

There are several limitations in present study that should be noted. First, although the median follow-up time was 70.7 months, more than half of the patients survived, and the number of events was insufficient, which may limit the statistical power of our findings. Second, despite the fact that we found associations between three SNPs and overall survival of GC, the mechanisms are still not clear and need to be further elucidated. Third, our study was based on a single group of patients. Hence, other independent replications and multi-centre studies need to be done to explore the role of genetic polymorphisms of lncRNA on 8q24 in the prognosis of GC in different populations.

## CONCLUSIONS

In summary, the present study revealed that the PRNCR1 rs12682421 AA genotype, the CASC8 rs1562430 TT genotype and the PCAT1 rs1026411 AA/AG genotype could serve as potential markers to predict the unfavourable survival of GC patients in the Chinese population. These three SNPs may be used as prognostic markers in combination with traditional clinical prognostic factors to refine therapeutic decisions for the individualised treatment of GC.

## ACKNOWLEDGEMENTS

We thank Ying Song for her work on follow-up of participants and the group of Gastric and Colorectal Surgery for their support.

### Funding

This work was supported by grants from the National Natural Science Foundation of China (81673145, 81703286, 81874279 and 81703293) and the Science and Technology Department of Jilin Province (20180414055GH). The funders had no role in study design, data collection and analysis, decision to publish, or preparation of the manuscript.

### Grant Disclosures

The following grant information was disclosed by the authors:
National Natural Science Foundation of China: 81673145, 81703286, 81874279 and 81703293.
Science and Technology Department of Jilin Province: 20180414055GH.

### Competing Interests

The authors declare that they have no competing interests.

### Author Contributions

- Yangyu Zhang conceived and designed the experiments, performed the experiments, analysed the data, prepared figures and/or tables, authored or reviewed drafts of the paper, and approved the final draft.
- Yanhua Wu conceived and designed the experiments, analysed the data, prepared figures and/or tables, authored or reviewed drafts of the paper, and approved the final draft.
- Zhifang Jia analysed the data, prepared figures and/or tables, authored or reviewed drafts of the paper, and approved the final draft.
- Donghui Cao performed the experiments, authored or reviewed drafts of the paper, and approved the final draft.
- Na Yang performed the experiments, analysed the data, prepared figures and/or tables, and approved the final draft.
- Yueqi Wang performed the experiments, analysed the data, authored or reviewed drafts of the paper, and approved the final draft.

- Xueyuan Cao conceived and designed the experiments, performed the experiments, authored or reviewed drafts of the paper, and approved the final draft.
- Jing Jiang conceived and designed the experiments, prepared figures and/or tables, and approved the final draft.

## Human Ethics

The following information was supplied relating to ethical approvals (i.e., approving body and any reference numbers):

The ethics committees of the First Hospital of Jilin University approved the study (2013-005).

## Data Availability

The raw measurements are available in the Supplemental Files.

## Supplemental Information

Supplemental information for this article can be found online at http://dx.doi.org/10.7717/peerj.8600#supplemental-information.

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
