# Peer review of "Long non-coding RNA polymorphisms on 8q24 are associated with the prognosis of gastric cancer in a Chinese population"

_PeerJ, doi:10.7717/peerj.8600_

## Round 0.1 · original submission · Major Revisions

Your manuscript has been reviewed and requires modifications prior to making a decision. The comments of the reviewer(s) are included at the bottom of this letter. Reviewer1 and Reviewer 3 indicated that the methods and discussion sections should be improved. Review 1 also has a concern about figures. I agree with the evaluation and I would, therefore, request for the manuscript to be revised accordingly.

Reviewer 1 ·

Basic reporting

In introduction section, the authors should clarify the therapy strategies of gastric cancer to emphasize the importance of the issue.

Experimental design

No comment.

Validity of the findings

1.In my opinion, the Fig.S1 should make a swap for Fig.2 to highlight the findings.
2. Type 1 error of multiple testing should be corrected.

Additional comments

This study shows some interesting findings regarding the prognostic biomarker signature for gastric cancer. The major issue is correlation between lncRNA SNPs and GC prognosis, the authors should emphasize the associated findings. In discussion section, at line 276-278, the prediction role of rs12682421 and rs1562430 should be discuss, and provide more information about correlation between therapy approaches and genetic background.

·

Basic reporting

Language is good and article is well organized and structured.

Experimental design

Study was done appropriate larger sample size. Research question, hypothesis and objectives were properly provided.

Validity of the findings

SNPs were well analyzed by using appropriate stats tool.

Additional comments

Overall good study as done on larger sample size and able to justify the hypothesis and objectives taken with proper results and discussion.

Reviewer 3 ·

Basic reporting

This study discusses the association between Inc-RNA polymorphisms and gastric cancer prognosis in a Chinese cohort. The manuscript is clear and unambiguous. Statistical analysis is comprehensively applied to clinical and experimental data. A few minor concerns and questions regarding the context are listed as following:

1. Line 107 and 108: Does “different human” mean different individuals or different races?

2. Since eight SNPs were analyzed in the same cohort, has linkage disequilibrium be considered in this study? If some of them are associated, influences from other loci need to be considered when analyzing the association between one SNP and GC risk or prognosis.

3. In Table 2, MSTs between different genotypes are not that significant, unfortunately. Even for rs12682421, the log-rank p-value is 0.03 without adjustment. Since statistical analysis involves multiple simultaneous statistical tests in this study, I highly recommend multiple test corrections to avoid type I error. It seems that rs12682421 has the most significant association with GC prognosis in Table 2. Have you further analysis which genotype has the closest relationship with GC survival, or which allele indicate better or worse prognosis?

4. More than two-thirds of references are before 2015, and only 10 references are published in recent 3 years. For innovation points, it’s better to introduce and discuss more recent works and publications.

Experimental design

It's already involved in basic reporting.

Validity of the findings

It's already involved in basic reporting.

Additional comments

It's already involved in basic reporting.

---

## Round 0.2 · accepted · Accept

Thank you very much for the submission of a revised version of your paper. I have gone through the track-changes manuscript and rebuttal letter and see that the authors addressed the reviewers' concerns and substantially improved the content of the manuscript. So, based on my own assessment as an academic editor, no further revisions are required, and the manuscript may be now accepted for publication in its current form.

Reviewer 1 ·

Basic reporting

no comment

Experimental design

no comment

Validity of the findings

no comment

Additional comments

no comment

Reviewer 3 ·

Basic reporting

Many thanks to the authors' responses. I don't have any new questions now.

Experimental design

No new comment.

Validity of the findings

No new comment.

Additional comments

No new comment.